# Natural Bioactive Epigallocatechin-Gallate Promote Bond Strength and Differentiation of Odontoblast-like Cells

**DOI:** 10.3390/biomimetics8010075

**Published:** 2023-02-10

**Authors:** Rene Garcia-Contreras, Patricia Alejandra Chavez-Granados, Carlos Alberto Jurado, Benjamin Aranda-Herrera, Kelvin I. Afrashtehfar, Hamid Nurrohman

**Affiliations:** 1Interdisciplinary Research Laboratory, Nanostructures, and Biomaterials Area, National School of Higher Studies (ENES) Leon, National Autonomous University of Mexico (UNAM), Leon 37684, Guanajuato, Mexico; 2Department of Prosthodontics, The University of Iowa College of Dentistry and Dental Clinics, Iowa City, IA 52242, USA; 3Clinical Sciences Department, College of Dentistry, Ajman University, Ajman City P.O. Box 346, United Arab Emirates; 4Department of Reconstructive Dentistry & Gerodontology, School of Dental Medicine, University of Bern, 3010 Bern, Switzerland; 5Missouri School of Dentistry & Oral Health, A. T. Still University, Kirksville, MO 63501, USA

**Keywords:** polyphenols, biomodification, dentin repair, resin–dentin bond

## Abstract

The (-)-*Epigallocatechin*-*gallate* (EGCG) metabolite is a natural polyphenol derived from green tea and is associated with antioxidant, biocompatible, and anti-inflammatory effects. Objective: To evaluate the effects of EGCG to promote the odontoblast-like cells differentiated from human dental pulp stem cells (hDPSCs); the antimicrobial effects on *Escherichia coli*, *Streptococcus mutans*, and *Staphylococcus aureus*; and improve the adhesion on enamel and dentin by shear bond strength (SBS) and the adhesive remnant index (ARI). Material and methods: hDSPCs were isolated from pulp tissue and immunologically characterized. EEGC dose-response viability was calculated by MTT assay. Odontoblast-like cells were differentiated from hDPSCs and tested for mineral deposition activity by alizarin red, Von Kossa, and collagen/vimentin staining. Antimicrobial assays were performed in the microdilution test. Demineralization of enamel and dentin in teeth was performed, and the adhesion was conducted by incorporating EGCG in an adhesive system and testing with SBS-ARI. The data were analyzed with normalized Shapiro–Wilks test and ANOVA post hoc Tukey test. Results: The hDPSCs were positive to CD105, CD90, and vimentin and negative to CD34. EGCG (3.12 µg/mL) accelerated the differentiation of odontoblast-like cells. *Streptococcus mutans* exhibited the highest susceptibility < *Staphylococcus aureus* < *Escherichia coli*. EGCG increased (*p* < 0.05) the dentin adhesion, and cohesive failure was the most frequent. Conclusion: (-)-*Epigallocatechin*-*gallate* is nontoxic, promotes differentiation into odontoblast-like cells, possesses an antibacterial effect, and increases dentin adhesion.

## 1. Introduction

Dental connective pulp tissue is located at the pulp chamber and the root canals. Its function is crucial for the regeneration after an injury enhances the odontoblast differentiation by dentinogenesis and DNA methylation [1]. The dentin is the main component and outer layer of the pulp chamber. Not only does dentin restore pulp, but odontoblasts also secrete it during their lifespan [2]. Dental caries can induce irreversible changes, such as necrosis of dental pulp tissue or at the odontoblast layers [3]. Bacterial adhesion and biofilm formation on dentin surfaces can result in the dissolution of mineralized tissue and decreased capacity for arresting caries progression. Several compounds, such as chlorhexidine, silver diamine fluoride, triclosan, and quaternary ammonium, have been investigated for their potential to inhibit biofilm formation on dentin surfaces [4,5]. Since the cytotoxic and collateral effects arise during the use of these compounds, in contrast, the natural metabolite bioactive component of green tea, (-)-*Epigallocatechin*-*gallate* (EGCG), has demonstrated their safe, biocompatible, anti-inflammatory, anticarcinogenic, antioxidant, antimicrobial, and remineralized dentin activity [6]. A variety of literature has demonstrated the potential use of EGCG for promoting the osteoblast and odontoblast differentiation of human mesenchymal dental pulp cells, as well as its ability to inhibit antibiofilm and promote the remineralization of dentin [7,8,9].

On the other hand, studies have shown that EGCG exhibits significant antibacterial properties through inhibiting virulence factors and modulation of glucosyltransferase-related gene expression. Additionally, EGCG has been shown to cause damage to bacterial cell membranes and inhibit the enzymes involved in fatty acid synthesis, potentially reducing the production of toxin metabolites. These findings indicate that EGCG may inhibit the formation of bacterial biofilms [6]. In addition to its potential antibiofilm properties, studies have also shown that green tea catechins have the ability to inhibit the activity of various bacterial enzymes, such as protein tyrosine phosphatase and cysteine proteinases in certain anaerobic oral bacteria, as well as DNA gyrase, and dihydrofolate reductase in bacteria and yeast [10]. EGCG has also been found to be effective against antibiotic-resistant strains of bacteria. Studies have demonstrated its efficacy in treating skin infections caused by multidrug-resistant Gram-negative bacteria such as *Pseudomonas aeruginosa* and *Escherichia coli*, which are known to be resistant to multiple antibiotics [11].

Furthermore, studies have demonstrated that EGCG possesses remineralization properties on dental tissues through tubule occlusion and collagen mineralization and improves the adhesion of bonding materials. As a natural extract derived from green tea, EGCG has been shown to prevent the growth of *S. mutans* biofilm and the degradation of dentin collagen through the inhibition of matrix metalloproteinase-2 (MMP-2) and MMP-9 expression [12,13]. The hybrid-layer degradation at the adhesive–dentin interface generated by adhesive hydrolysis, collagen enzymolysis by MMPs and cysteine cathepsin, and secondary caries are the main factors contributing to the failure of bonding stability [14].

While previous studies have indicated that EGCG may stimulate odontoblastic differentiation, suppress bacterial growth, and enhance adhesion of enamel and dentin, it is important to investigate further the effects of EGCG on other biological properties of human dental pulp stem cells (hDPSCs) in addition to its antibacterial and remineralization properties before it can be considered for use in vital pulp therapy and treatment. Here, the study hypothesized the use of EGCG as a biomimetic protective compound with potential differentiation of hDPSCs to odontoblasts, antibacterial activity, and improvement of the bond strength on enamel and dentin. The purpose of this study was to evaluate the potential use of EGCG to promote in vitro odontoblast-like differentiation of hDSPCs, promote antibacterial activity against *Escherichia coli*, *Streptococcus mutans*, and *Staphylococcus aureus*, and improve adhesion to enamel and dentin by shear bonds strength (SBS) and adhesive remnant index (ARI).

## 2. Materials and Methods

### 2.1. hDPSC Cell Culture

The protocol for the isolation, culture, and characterization of hDPSCs was evaluated by the bioethics committee of the ENES Leon Unit, UNAM, for authorization with the registration code CE_16 004_SN. We used permanent erupted third molars indicated for odontectomy in the ENES Leon Unit clinics of patients aged 16–25, free of pulpal and periapical pathology. The pulp tissue was isolated inside the horizontal laminar flow hood. The molar was cut at the level of the coronal–root junction with a low-speed turbine with a carbide disc with constant cooling until close to the pulp chamber. The pulp tissue was obtained and 1 × 1 mm explants were performed and incubated with minimum essential medium eagle medium (MEM, Sigma-Aldrich, Saint Louis, MO, USA) added with 20% fetal bovine serum (FBS, Gibco, USA), 1% glutamine (Gibco), and 1% antibiotic (Sigma-Aldrich) in Petri dishes 10 cm in diameter and incubated at 37 °C with 5% CO_2_ and 95% humidity for 21 days

### 2.2. Cell Characterization

The hDPSCs showed a fibroblastoid morphology and a confluence of 90%, and subcultures were cultivated with MEM medium added with 10% fetal bovine serum, glutamine, and antibiotics. The characterization of the cells was carried out with a cellular passage of four cell divisions (4 population doubling level (PDL)) following the reported criteria, adherence to plastic, and fibroblastoid morphology; and characterized with flow cytometry (Thermo-Scientific, Waltham, MA, USA) anti-human-CD90 superbright 436 and anti-human-CD105 Alexa Fluor (Sigma-Aldrich), and with immunocytochemistry to vimentin CD34 (Sigma-Aldrich).

### 2.3. hDPSC–EGCG Dose Response

The hDPSCs were subcultured (4 PDL) in a 96-microwell plate. Cells were washed with PBS (–) and detached by 0.25% trypsin–0.025% EDTA-2Na in PBS (–) (Sigma-Aldrich) for each experiment. The number of inoculated cells was determined by excluding trypan blue with a hemocytometer under light microscopy. A total of 1 x10^5^ cells/mL were subcultured for 48 h to allow complete attachment. The cells were treated with EGCG (Sigma-Aldrich, 0–25 µg/mL, *n* = 9), then incubated for 24 h with fresh culture medium. The cell viability was determined by the MTT method. In brief, cells were incubated for 7 h with 0.2 mg/mL (Thiazolyl Blue Tetrazolium Bromide, Sigma-Aldrich) in fresh MEM with 10% FBS. The formazan formed during incubation was dissolved with 0.1 mL of dimethyl sulfoxide (DMSO, Karal, Mexico). The absorbance at 570 nm of the lysate was determined by using a microplate spectrophotometer reader (Multiskan go, Thermo-Scientific, Helsinki, Finland). The cytotoxicity was based on the ISO 10993-5:2009 Biological evaluation of medical devices—Part 5: Tests for in vitro cytotoxicity.

### 2.4. Odontoblast-Like Differentiation of hDPSC

To promote odontoblast-like differentiation, hDPSCs (6 PDL) were treated with odontoblastic differentiation medium: MEM, 10% FBS, 1% antibiotics, 0.1 mM of dexamethasone (Sigma-Aldrich), 5 mM of β-glycerophosphate (Sigma-Aldrich), 50 mg/mL of ascorbic acid (Sigma-Aldrich), 20 ng/mL of transforming growth factor-β3 (Sigma-Aldrich), and 5 ng/mL of fibroblast growth factor-2 (Sigma-Aldrich), enriched with 3.12 µg/mL of EGCG for 10 days. Conventional subculture cells were used as control. Mineral deposition activity was evaluated by alizarin red stain, Von Kossa stain, and collagen/vimentin (Masson trichrome stain, Sigma-Aldrich). To evaluate calcified minerals, the alizarin red was dissolved for 16 h with 5% 2-isopropanol (Sigma-Aldrich) and 10% acetic acid solution (Sigma-Aldrich). Absorbance was at 550 nm in a microplate reader spectrophotometer [15,16]. The calcium deposition was evaluated by performing a Von Kossa stain using a 5% silver nitrate (Sigma-Aldrich) solution under ultraviolet light for 20 min, followed by a 5% sodium thiosulphate (Sigma-Aldrich) solution for 5 min. Then, counterstaining with a nuclear fast red solution (Sigma-Aldrich) for 10 min, the nucleus was stained black. To evaluate the collagen and vimentin expression in the cells, the cultures were fixed with formaldehyde and treated with collagen/vimentin antibodies. The resulting staining of all methods was photographed for analysis.

### 2.5. Antimicrobial Test

To perform the broth microdilution assays, three certified bacterial strains were used: *Escherichia coli* (ATCC 8739), *Streptococcus mutans* (ATCC 36668), and *Staphylococcus aureus* (ATCC 6538). For the broth microdilution assay, the Mueller Hinton growth medium (Sigma-Aldrich) was prepared per the manufacturer’s instructions. Then, the bacteria were placed on Mueller Hinton agar plates (Mueller Hinton BD Bioxon, Mexico) and kept at 37 °C for 24 h for incubation. The same methodology was followed with each bacterium. Briefly, 6 homogeneous colonies were placed in 15 mL of Mueller Hinton broth; this suspension was incubated for 24 h at 37 °C. The bacteria were cultivated at 0.5 on the McFarland scale (Containing about 1 × 10^8^ (CFU)/mL), then 30 µL was used to prepare a working solution at a final dilution of 1:1000. The EGCG was inoculated at different concentrations from 0 to 4 µg/mL and incubated for a further 24 h at 37 °C. The number of surviving bacteria was determined by 0.2 mg/mL of MTT (Thiazolyl Blue Tetrazolium Bromide, Sigma-Aldrich), which was incubated in the dark for 4 h at room temperature. The microplate was analyzed with a microplate spectrophotometer reader (Thermo-Scientific) at 595 nm. For the internal validity design, the sterile saline solution was used as the negative control, with 0.12% chlorhexidine hydrate (Dentscare Ltd.a, FGM, Brazil) as the positive control. Assays were performed in triplicate from three independent experiments (*n* = 9). The data were analyzed following the Clinical and Laboratory Standards Institute (CLSI).

### 2.6. Demineralization and Samples Preparation

Forty human teeth (*n* = 40) were freshly extracted. Half of the teeth had the middle surface of the dentin exposed. The samples were ultrasonically cleaned in distilled water for 5 min. Then, samples were immersed in demineralizing solution (1.5 mM of CaCl_2_, 0.9 mM of KH_2_PO_4_, 150 mM of KCl, and 0.1 mM of sodium acetate at pH 4.5) and incubated at 37 °C with 180 rpm for 360 h. The demineralization evaluation was conducted with a DIAGNOdent pen (Kavo-Kerr, Berlin, Germany), as previously reported in our group’s research [17]. The solution was replaced every 48 h. Then, the samples were mounted in cylindrical acrylic blocks, according to ISO 29022:2013 Dentistry–Adhesion–Notched-edge shear bond strength test. A standardized metallic device was used to make composite blocks of 4 × 4 × 1 mm (Ivoclar Vivadent, Te-Econom plus, Shann/Liechtenstein). Samples were randomly grouped into four groups (*n* = 10) as follows: 1: dentin SBS; 2: dentin SBS with 3.12 µg/mL of EGCG; 3: enamel SBS; and 4: enamel SBS with 3.12 µg/mL of EGCG.

The dentin and enamel surfaces were prepared using a total-etch approach per the instructions provided by the manufacturer. The preparation involved the application of 37% orthophosphoric acid for 15 s on dentin and 30 s on enamel. Groups 1 and 2 were treated with two coats of the adhesive system (Ivoclar Vivadent, Te-Econom Bond, Shann/Liechtenstein), applied for 10 s each, and were air-dried and light-cured for 20 s with a BlueLex LD-105 device (New Taipei City, Taiwan) with an intensity of >1000 mW/cm^2^. For groups 3 and 4, a cross-linking agent, EGCG, was added to the same adhesive system, in a proportion of 1:4, respectively (3.12 µg/mL). The samples were then vortexed for 1 min before being applied to the enamel and dentin surfaces. A composite resin was then applied to the composite block’s base and placed under gentle pressure until it reached the surface. Resin excess was removed with a microbrush, and light-curing was performed by placing the device perpendicular to the external surface of the resin block for 30 s.

### 2.7. Shear Bond Strength (SBS) and Adhesive Remnant Index (ARI)

Before testing, the samples were immersed in distilled water for 48 h at 37 °C and a relative humidity of 100%. To measure the shear bond strength (SBS), a universal testing machine (Mecmesin, advanced force/torque indicator (AFTI), London, UK) was used. The equipment applied a stainless-steel blade of 10 mm in length to the interface between the tooth and composite at a cross speed of 1 mm/minute until failure occurred. The SBS was determined in megapascals (MPa) by dividing the maximum load (measured in Newtons) by the area of the composite block. The amount of residual resin remaining on the enamel and dentin was assessed using an optical stereomicroscope with ×20 magnification and classified according to the adhesive remnant index (ARI) as follows: 0 = No adhesive or composite present on enamel or dentin; 1 = Less than 50% of the adhesive or composite left on enamel or dentin; 2 = More than 50% of adhesive or composite left on enamel or dentin; 3 = All of the adhesive or composite remaining on enamel or dentin.

### 2.8. Statistical Analysis

The data was expressed as mean ± standard deviation and was analyzed with the Shapiro–Wilks normality test and ANOVA post hoc Tukey-test. A significance level of *p* < 0.05 and a confidence interval of 95% were used. The results of the adhesive remnant index (ARI) were analyzed with the χ^2^ test.

## 3. Results

### 3.1. hDPSC Characterization

Primary hDPSC culture characterization was positive to CD90 and CD105 in flow cytometry (Figure 1) and vimentin and negative to CD34 with immunohistochemistry.

The population of cells labeled with the anti-CD90 antibody and anti-CD105 antibody corresponded to about 60% and 59%, respectively. In orange, the population of negative cells (autofluorescence), and in blue, positive cells (Figure 1A–D). The population of double-positive cells for CD105 (super bright 436) and CD90 (Alexa Fluor) corresponded to 49.9%. Figure 2 shows the microphotographs obtained after the immunocytochemistry showed that the vimentin was strongly positive (Figure 2A,B) and negative for CD34 (Figure 2C,D). The mesenchymal origin of the hDPSC primary cell culture was confirmed.

### 3.2. hDPSC-EGCG Dose-Response

Near-confluent hDPSCs in contact with different doses, after 24 h, exhibited no cytotoxic effects in a range of 0–12.5 µg/mL; meanwhile, the use of 25 µg/mL reduces significantly (*p* < 0.05) the viable cell number in a dose-dependent manner at 58 ± 9.2%. Figure 3 shows the dose response of EGCG in culture with hDPSCs.

### 3.3. Odontoblast-Like Differentiation of hDPSCs

Figure 4 shows the alizarin red, Von Kossa, and collagen (Masson trichrome stain)/vimentin immunochemistry. In the control group, in which conventional growth media was used (Figure 4A,D,G and Figure 5), no mineral deposited on the cell surface was observed, and only a minimal amount of collagen/vimentin was present.

In the odontoblast differentiation medium group without EGCG, a small amount of matrix mineralization and calcified nodes with collagen/vimentin were observed (Figure 4B,E,H and Figure 5, *p* < 0.05). In contrast, the medium containing EGCG 3.12 µg/mL (Figure 4C,F,I and Figure 5, *p* < 0.01)) exhibited statistically significantly more matrix mineralization and calcified nodes with stains when compared to the control group and without EGCG. Although not enough mineral deposition was observed due to the short culture period of the cells, mineral deposition was formed along the cytoplasm and was observed on the cell surface.

### 3.4. Antibacterial Activity

Figure 6 corresponds to the results obtained in the antimicrobial tests using EGCG through microdilution in broth. *Streptococcus mutans, Staphylococcus aureus* (*Gram*-*positive* type, facultative anaerobe), and *Escherichia coli* (*Gram*-*negative* bacillus, facultative anaerobe). The results show that EGCG influences growth inhibition significantly compared to the control group (chlorhexidine 0.12%). The bacteria with the highest susceptibility to EGCG were *Streptococcus mutans* < Staphylococcus aureus < *Escherichia coli*. Regarding the effect of EGCG on *Streptococcus mutans*, there was a nonsignificant (*p* > 0.05%) effect with respect to chlorhexidine at the concentration of EGCG 3.12 µg/mL, with an MIC of 1 µg/mL, being highly sensitive. In the case of *Staphylococcus aureus*, there was a nonsignificant difference at the 0.5% concentration, with an MIC of 1 µg/mL. It was similarly sensitive. The results showed that the bacterium with the greatest resistance was *Escherichia coli* since the inhibitory effect of EGCG was reached at the maximum concentration (4 µg/mL), with an MIC at the same concentration. It might be thought that from the data obtained, the effect on growth inhibition could be dose-dependent. Finally, *Streptococcus mutans* exhibited the highest susceptibility to EGCG (MIC50, 0.5–1 µg/mL and MIC90, 1–2 µg/mL) *Staphylococcus aureus and Escherichia coli* (MIC50, 1–3 µg/mL and MIC90, 2–4 µg/mL).

### 3.5. Shear Bond Strength (SBS) and Adhesive Remanent Index (ARI)

Figure 7 represents the results of the SBS from the four analyzed groups. The presence of EGCG at 3.12 µg/mL significantly enhances (*p* < 0.01) the SBS in dentin compared with the control (17.5 ± 1.3 MPa and 15.1 ± 2.1 MPa, respectively). Meanwhile, there was no difference (*p* > 0.05) in the enamel groups. The ARI did not exhibit any differences (*p* > 0.05) between control groups. The most frequent failure was (50%) cohesive failure in the groups with EGCG (60%).

## 4. Discussion

### 4.1. Dose Response and Odontoblast-Like Differentiation of hDPSCs

The EGCG has been used in contact with hDPSCs at 10 µg/mL without altering the cell viability nor significant cytotoxic effect [8], similar to the used doses in this study with a range of 0–25 µg/mL. Results showed that EGCG inhibited hDPSC proliferation dose-dependently with concentrations above 12.5 µg/mL (Figure 3). Previous studies have reported varying effects of EGCG on cell proliferation, with some findings that high doses (3.7–4.6 µg/mL) of EGCG were cytotoxic and inhibitory to human periodontal ligament cells, while lower doses showed no significant effect [18]. In this study, a concentration of 3.12 µg/mL was used to promote hDPSC proliferation and differentiation into odontoblast-like cells (Figure 3, Figure 4 and Figure 5), and the amount of calcified deposits dissolved by alizarin red evaluation was observed to be statistically significantly higher (*p* < 0.01).

The development of mineralized tissues and the differentiation of stem cells into odontoblasts play an essential role in the regeneration of dental hard tissues. In response to injury, dental pulp cells differentiate into odontoblasts and form reparative dentin by building on the pulp–dentin complex. Many studies have examined the use of various compounds to differentiate hDPSCs in vitro. The combination of silica-based compounds with EGCG has been shown to enhance the differentiation of hDPSCs by increasing the ratio of calcium to phosphorus, promoting mineralization of collagen forming a calcified matrix, and increasing the activity of the enzyme alkaline phosphatase (ALP) after 7 days of culture [9]. Our findings support these conclusions, as we observed that the presence of EGCG led to an increase in mineral deposition and the presence of collagen in hDPSCs after 10 days of culture, as shown in Figure 3, Figure 4 and Figure 5.

In future experiments, it is recommended to evaluate the expression of genes associated with osteo/odontogenic differentiation, such as ALP, RunX2, OSX, OCN, DMP-1, and DSPP, as well as the ratio of calcium to phosphorus (Ca^+^/P) in the presence of EGCG. This will provide further insight into how EGCG promotes mineral deposition and collagen expression during the differentiation of hDPSCs into odontoblast-like cells. In addition, studying the binding of calcium ions to the plasma membrane and the potential impact on stem cell migration may also provide valuable information on the cellular mechanisms involved in odontogenesis.

### 4.2. Antibacterial Activity

Chlorhexidine, a common antimicrobial agent, is often used in various oral-health treatments and procedures. However, prolonged chlorhexidine use has been linked to several side effects, such as oral irritation, discoloration of tissues, and change in taste perception. Therefore, there is a growing interest in identifying natural alternatives with minimal side effects that can serve as effective alternatives. Figure 6 shows that EGCG significantly reduces the growth of *Streptococcus mutans*, *Staphylococcus aureus*, and *Escherichia coli* when compared to the control group (0.12% chlorhexidine). Previous studies have also reported the ability of EGCG to reduce bacterial populations in the oral cavity. Notably, EGCG has been found to inhibit specific genes involved in biofilm formation, such as gtf B, C, and D, without significantly impacting the overall growth of oral bacteria [19]. This makes EGCG a promising natural antiplaque agent. Recently, EGCG has been demonstrated to inhibit bacterial adherence to hard surfaces by affecting cell-surface hydrophobicity and charge. This metabolite induces the formation of holes in the cell surfaces of *Streptococcus mutans* without impacting their survival or the development of biofilms [20]. This last finding is supported by in vivo studies, wherein the use of EGCG solution as a rinse resulted in a decrease in the amount of *Streptococcus mutans* and *lactobacilli* present in the oral cavity of children [21].

Additionally, previous research has shown that the antibacterial activity of EGCG on *Escherichia coli* is attributed to an enhancement in intracellular reactive oxygen species (ROS) and a weakened adaptive oxidative stress response [22]. In addition, previous studies have shown that polyphenols, including EGCG, can inhibit the ATP synthase enzyme in *Escherichia coli* [23]. Research has explored the effects of EGCG in combination with oxytetracycline against strains of *Staphylococcus aureus*, including antibiotic-resistant strains. The results of Novy et al. (2013) showed a solid synergistic antimicrobial against *Staphylococcus aureus* and *Escherichia Coli* when combined with the polymer chitosan, which aligns with the findings of the current study [24].

### 4.3. EGCG Improves Adhesion

The proteolytic activity in the hybrid layer directly impacts the longevity of the resin-based restoration. This hybrid-layer degradation is caused by enzymes from the zymogen protease family in the extracellular matrix, specifically matrix metalloproteinases (MMPs) and cysteine cathepsins (CCs). These enzymes are activated in an acidic environment, such as acid-etching conditioning (phosphoric acid) or adhesive itself [25]. Pretreatment with chlorhexidine after acid conditioning can reduce the activity of the zymogen protease family. However, it is associated with cytotoxic and genotoxic effects [26,27,28], limiting its clinical use as the gold standard in adhesive dentistry.

The use of EGCG as a preconditioning agent in dental adhesives has been investigated in recent studies. The results of these studies have shown that EGCG does not significantly affect the bond strength of the resin-based restoration in affected dentin, with bond strengths of 14.2 ± 4.6 MPa for the control group and 0.1% and 14.7 ± 3.8 MPa for the EGCG group. However, when EGCG is used as a preconditioning agent in sound dentin, it has been found to increase microtensile bond strengths of 12.6 ± 3.4 MPa at a concentration of 0.1% and a lesser bond strength at a concentration of 0.5% (5.2 ± 2.9 MPa) [29]. Our results, as illustrated in Figure 7, indicate that using EGCG in the adhesive resulted in a bond strength of 17.5 ± 1.3 MPa, while the adhesive without EGCG had a bond strength of 15.1 ± 2.1 MPa. These results are slightly higher than those reported previously. However, they are comparable to the findings by Zhang et al. (2022), which reported a bond strength of 17.3 ± 1.34 MPa for dentin treated with 0.015 EGCG and 14.4 ± 1.70 MPa for the control dentin [30]. Our findings align with previous studies that suggest that using EGCG may inhibit collagenase degradation by MMP-2 and MMP-9. Additionally, the formation of inter- and intramolecular cross-links through the presence of EGCG enhances the mechanical properties and resistance to enzymatic degradation of the collagen matrix [29,31].

## 5. Conclusions

These results suggest that (-)-*Epigallocatechin*-*gallate* is not cytotoxic in culture with human dental pulp cells (hDPSCs). The presence of 3.12 µg/mL of EGCG promotes the differentiation of the odontoblast-like cells and possesses an antibacterial effect against *Streptococcus mutans*, *Staphylococcus aureus*, and *Escherichia coli*, and incorporation into adhesive improves the resin composite adherence to dentin.

## Figures and Tables

**Figure 1 biomimetics-08-00075-f001:**
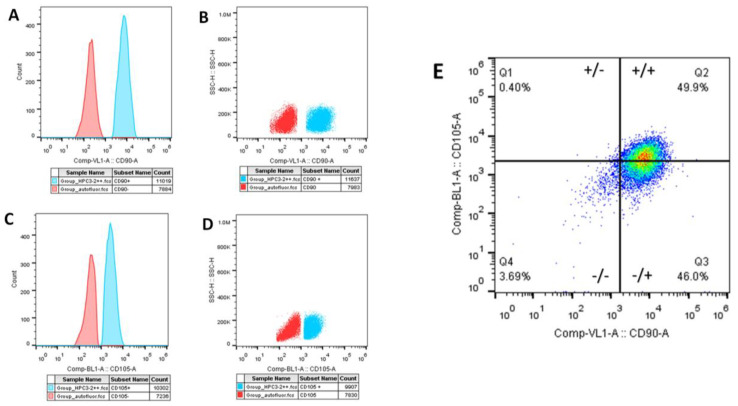
Flow cytometric characterization of hDPSCs. hDPSCs (90% confluence, 4 PDL) were cultivated for 48 h. (**A**,**B**). The population of cells labeled with the antibody anti-CD90 was 60% positive (**C**,**D**), and the population labeled with the anti-CD105 antibody was 59% positive. In orange, the population of negative cells (autofluorescence) and in blue, the positive cells. (**E**) Population of double-positive cells for CD105 and CD90, corresponding to 49.9% positive. Antibody used anti-human-CD90 superbright 436 and anti-human-CD105 Alexa Fluor. hDPSCs = human dental pulp stem cells; PDL = population doubling level.

**Figure 2 biomimetics-08-00075-f002:**
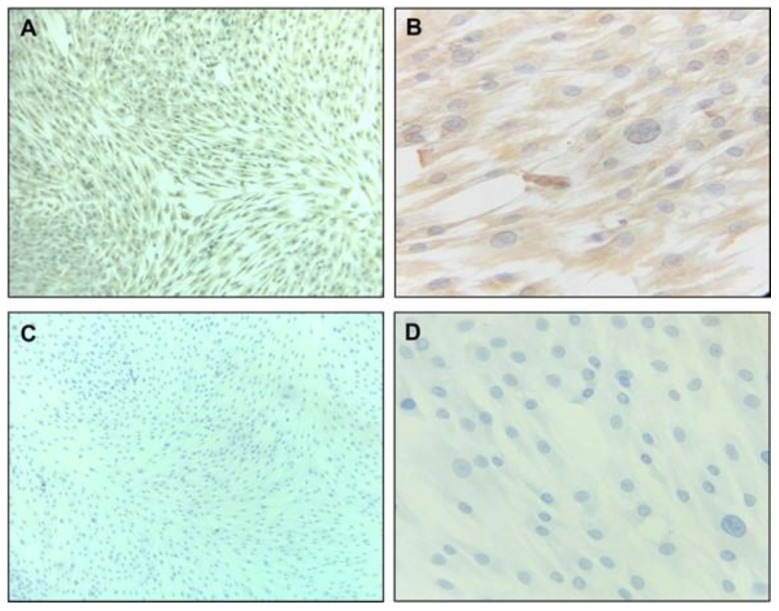
Immunohistochemistry characterization of hDPSC primary culture. hDPSCs (90% confluence, 4 PDL) were cultivated for 48 h. Microphotographs (**A**,**B**) show strong, positive vimentin, and (**C**,**D**) show negative for CD34, at 20× and 40×, respectively. hDPSCs = human dental pulp stem cells; PDL = population doubling level.

**Figure 3 biomimetics-08-00075-f003:**
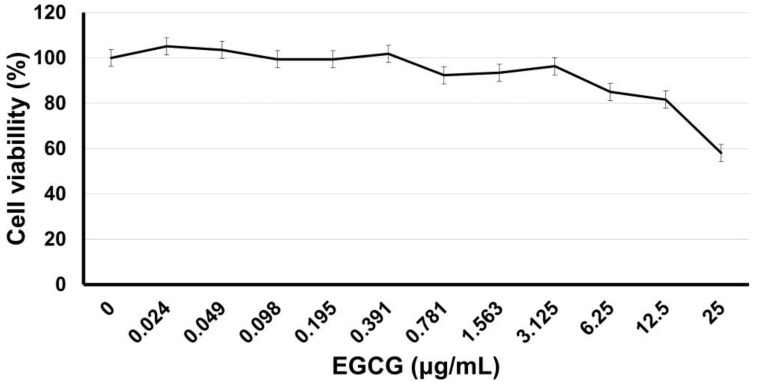
EGCG dose response in culture with hDPSCs. Near confluent hDPSCs (90% confluence, 4 PDL) were incubated for 24 h with EGCG at different concentrations from 0 to 25 µg/mL. After incubation, the relative viable cell number was determined by MTT assay. Each value represents the Mean ± SD of triplicate assays (*n* = 9), 570 nm absorbances (0.247; 0.527). EGCG = (-)-*Epigallocatechin*-*Gallate*; hDPSCs = human dental pulp stem cells; PDL = population doubling level; MTT = 3-[4,5-dimethylthiazol-2yl]-2,5-diphenyltetrazolium bromide; SD = standard deviation.

**Figure 4 biomimetics-08-00075-f004:**
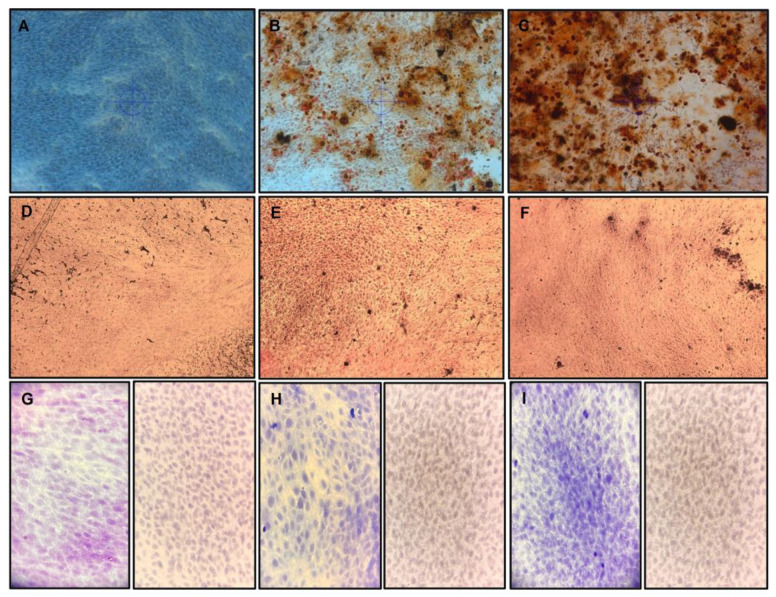
hDPSC differentiation to odontoblast-like cells in culture with or without EGCG for 10 days. hDPSCs (100% confluence, 6 PDL) were subcultured, differentiated to odontoblast-like cells, and evaluated by alizarin red stain (**A**–**C**), Von Kossa stain (**D**–**F**), and immunohistochemistry (**G**–**I**); Left = Masson trichrome stain, right = Vimentin antibody. (**A**,**D**,**G**) hDPSC controls were incubated with conventional culture medium; the hDPSCs did not exhibit matrix mineralization, and a small number of calcified nodes with collagen/vimentin. (**B**,**E**,**H**) hDPSCs were treated with the odontoblastic medium without EGCG, a small amount of matrix mineralization, and calcified nodes with collagen/vimentin. (**C**,**F**,**I**) hDPSCs were treated with odontoblastic medium containing 3.12 µg/mL, exhibiting more matrix mineralization and more calcified nodes with Masson trichrome stain collagen/vimentin. All microphotographs were obtained under a light microscope at 20×. EGCG = (-)-*Epigallocatechin*-*Gallate*; hDPSCs = human dental pulp stem cells.

**Figure 5 biomimetics-08-00075-f005:**
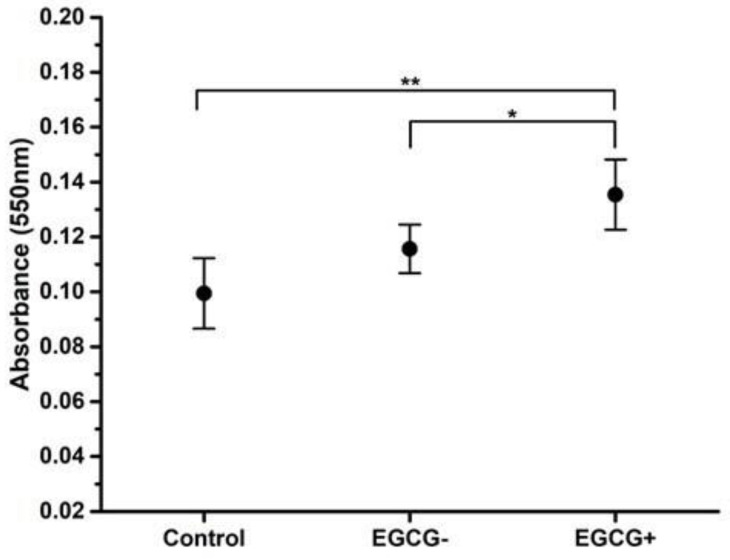
Alizarin red calcified deposits dissolved of hDPSC differentiation to odontoblast-like cells. hDPSCs (100% confluence, 6 PDL) were subcultured and promoted to odontoblastic-like differentiation control, with or without medium containing 3.12 µg/mL of EGCG for 10 days. The calcified deposits after alizarin red stains were dissolved with extraction of 5% 2-isopropanol and 10% acetic acid solution for 16 h. Each value represents the mean ± SD of triplicate assays (*n* = 24), * *p* < 0.05, ** *p* < 0.01 ANOVA post hoc Tukey test. Absorbances, 550 nm (0.061; 0.1675). EGCG = (-)-*Epigallocatechin*-*Gallate*; hDPSCs = human dental pulp stem cells; PDL = population doubling level; SD = standard deviation.

**Figure 6 biomimetics-08-00075-f006:**
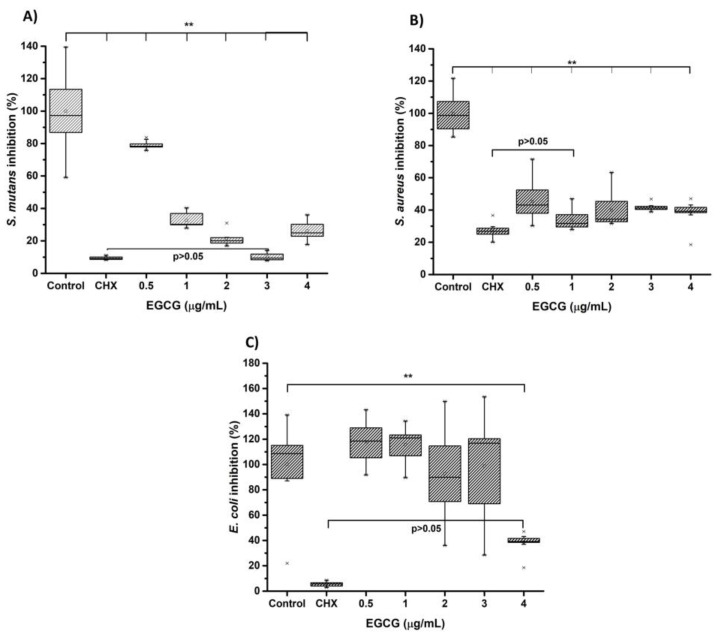
Antibacterial activity of EGCG with (**A**) *Streptococcus mutans* (ATCC 36668), (**B**) *Staphylococcus aureus* (ATCC 6538, and (**C**) *Escherichia coli* (ATCC 8739). The bacteria were cultivated at 0.5 on the McFarland scale. The EGCG was inoculated at different concentrations from 0 to 4 µg/mL and incubated for 24 h at 37 °C. The number of surviving bacteria was determined by MTT assay. Each value represents the mean ± SD of triplicate assays (*n* = 9), ** *p* < 0.01 ANOVA post hoc Tukey test. Absorbances, 595 nm. EGCG = (-)-*Epigallocatechin*-*Gallate*; ATCC = American type of cell culture; MTT = 3-[4,5-dimethylthiazol-2yl]-2,5-diphenyltetrazolium bromide; CHX = chlorhexidine, SD = standard deviation.

**Figure 7 biomimetics-08-00075-f007:**
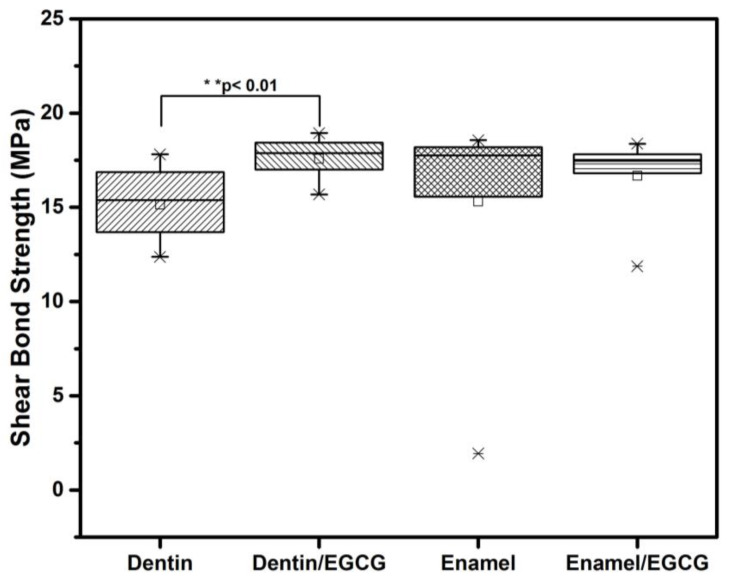
EGCG remineralization and shear bond strength (SBS). The enamel and dentin were demineralized with a solution containing 1.5 mM of CaCl_2_, 0.9 mM of KH_2_PO_4_, 150 mM of KCl, and 0.1 mM of sodium acetate at pH 4.5 for 360 h at 37 °C with 180 rpm. Standardized resin blocks of 4 × 4 × 1 mm adhered to enamel and dentin as follows: Group 1: dentin SBS; Group 2: dentin SBS with EGCG 3.12 µg/mL, Group 3: enamel SBS; Group 4: enamel SBS with EGCG 3.12 µg/mL. After 2 days in distilled water at 37 °C, the SBS was performed at a cross speed of 1 mm/min. Each value represents the mean ± SD megapascals (MPa) of *n* = 10, ** *p* < 0.01 ANOVA post hoc Tukey test. EGCG = (-)-*Epigallocatechin*-*Gallate*; SD = standard deviation.

## Data Availability

Not applicable.

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
