# Peer review of "Natural Bioactive Epigallocatechin-Gallate Promote Bond Strength and Differentiation of Odontoblast-like Cells"

_biomimetics, 2023, doi:10.3390/biomimetics8010075_

Round 1

Reviewer 1 Report

The manuscript “Natural Bioactive Epigallocatechin-Gallate induces Remineralization and Differentiation of Odontoblast-like Cells” is an interesting study. These are some of my comments for improvement:

1.     The format of some expression should be consistent: for example, line 287, line 303, line 416-417: “S. aureus”, “S. mutans” have to be reported in italic. Bacteria names have been reported sometimes in the full form, sometime with the abbreviation. Line 281: “Day 10” should be “day 10”. Line 402: “.5 %” should be corrected.

2.     The introduction could be improved to better present the topic in a more explanatory context, highlighting what could be the novelty of the work.

3.     References are not reported in a correct format. Please, correct them. 

4.     In Materials and methods section, some references about the protocols and sample preparation are missing. 

5.     Figure 1 has a very low resolution, the table subscription is not visible. The letters (A,B,C,D,E) are not indicated. 

6.     In the discussion, lines 336-241, an explanation on the choice of the EGCG concentration of 3.12 μg/mL is lacking. The references should be inserted to explain this choice.

7.     Line 386, the sentence seems not to be finished “..this metabolite has also been associated with polymers such as chitosan, where strong antimicrobial activity against S. aureus and E. coli has also been observed coinciding with our results in which [24].”

Author Response

  1. The format of some expression should be consistent: for example, line 287, line 303, line 416-417: “S. aureus”, “S. mutans” have to be reported in italic. Bacteria names have been reported sometimes in the full form, sometime with the abbreviation. Line 281: “Day 10” should be “day 10”. Line 402: “.5 %” should be corrected.

Answer: Thank you for the feedback. The suggestion was corrected, and the bacteria were reported constantly in line 33, 301, 302, 305, 306, 308, 311, 314, 315, 381, 393, 396, 398, 401, 404, 438, 439.

In line 402; the text was rephrased and the term was omitted.

Line 402 change for line424 and the suggestion was corrected.

  1. The introduction could be improved to better present the topic in a more explanatory context, highlighting what could be the novelty of the work.

Answer: The background information was modify in order to be more explanatory, and the goal of the study was described in the introduction.

  1. References are not reported in a correct format. Please, correct them. 

Answer: Thank you. The format of the references was made uniform through correction.

  1. In Materials and methods section, some references about the protocols and sample preparation are missing. 

Answer: The material and methods section include relevant references, while some sections featured original methodology.

  1. Figure 1 has a very low resolution; the table subscription is not visible. The letters (A,B,C,D,E) are not indicated. 

Answer: We have included the letters A, B, C, D in Figure 1 . Unfortunately, the clarity of the text in the boxes may not be optimal due to its origin from flow cytometry, but the figure legends provide a clear understanding of the content.

  1. In the discussion, lines 336-241, an explanation on the choice of the EGCG concentration of 3.12 μg/mL is lacking. The references should be inserted to explain this choice.

Answer: The dose was determined through a dose-response assay. The text in the discussion section from lines 348 to 357 was improved for better clarity.

  1. Line 386, the sentence seems not to be finished “..this metabolite has also been associated with polymers such as chitosan, where strong antimicrobial activity against S. aureus and E. coli has also been observed coinciding with our results in which [24].”

Answer: The text for lines 403-405: was improved The results, as reported by Novy et al. (2013), showed a strong synergistic antimicrobial against Staphylococcus aureus and Escherichia coli when combined with the polymer chitosan, which aligns with the findings of the current study[24].

Reviewer 2 Report

The manuscript is well written and developed. The methodology used is rigorous and controls have been included to increase internal validity.

Previously validated methods are used.

The results are clear and properly discussed, with concrete conclusions.

References follow the required format.

I recommend its publication without changes, just make a fine revision of the writing of the English language.

Author Response

Thank you for the feedback.

The updated version of the manuscript was revised and improved by a native English speaker.

Reviewer 3 Report

The ms. by Jurado  and coworkers presents an experimental study on the role of Epigallocatechin-gallate (EGCG) in odontoblast like differentiation of pulp stem cells, antimicrobial activity towards three different bacterial strains and shear bond strength involving dentin and enamel. The ms. presents some interesting data and should be considered for publication. However, it should be significantly reworded and reorganized, in its present form it is difficult to follow, it contains some overstatements or lack of information.

Here some main points:

Remineralization: I do not see in the ms. any experiment addressing remineralization. I see an experiment about differentiation of pulp stem cells, another involving antimicrobial activity, a third experiment presenting data on shear bond strength of a dental adhesive to dentin and enamel. In the abstract it is written: “remineralization was conducted by incorporating EGCG in an adhesive system and tested with SBS-ARI”. This is wrong and confusing. This is not an experiment on remineralization, no mineral, hard tissue is deposited in this experiment, there are no cells in the tested samples that could deposit it. Rather, this is an interesting experiment on the effect of EGCG on SBS of a commercial adhesive. As reported in the literature, the observed effect is likely due to crosslinking of dentin proteins of the hybrid layer and ensuing strengthening of the same. By the way, the Authors should indicate in the experimental section how EGCG is applied in the present case. As a solid? Or in solution? Of what?

Differentiation of pulp stem cells. It is claimed, in the title and throughout  the ms., that EGCG induces odontoblast-like differentiation of pulp stem cells. This is not what is reported in the paper. Differentiation of pulp stem cells is induced by the differentiating medium; EGCG apparently magnifies the effect of the differentiating medium. To claim that EGCG induces differentiation, tests should be conducted with EGCG without differentiating medium. Again, the magnifying effect of EGCG is of interest, the claim is wrong.

The conclusions of the abstract and of the paper are, in this reviewer views, correct , reasonable and of interest: EGCG PROMOTES (this is different from INDUCES) odontoblast-like differentiation of pulp stem cells, possesses antibacterial effect, and increase SBS of dentin to the adhesive. The Authors should reword the ms. sticking to these sounds conclusions, not invoking mechanisms of phenomena, such as remineralization, that are not addressed in their experiments

Author Response

Remineralization: I do not see in the ms. any experiment addressing remineralization. I see an experiment about differentiation of pulp stem cells, another involving antimicrobial activity, a third experiment presenting data on shear bond strength of a dental adhesive to dentin and enamel. In the abstract it is written: “remineralization was conducted by incorporating EGCG in an adhesive system and tested with SBS-ARI”. This is wrong and confusing. This is not an experiment on remineralization, no mineral, hard tissue is deposited in this experiment, there are no cells in the tested samples that could deposit it. Rather, this is an interesting experiment on the effect of EGCG on SBS of a commercial adhesive. As reported in the literature, the observed effect is likely due to crosslinking of dentin proteins of the hybrid layer and ensuing strengthening of the same. By the way, the Authors should indicate in the experimental section how EGCG is applied in the present case. As a solid? Or in solution? Of what?

Answer: Thank you for your revision. The title was changed to accurately represent the experiment performed. The Materials and Methods section was updated to include the EGCG concentration, determined through a dose-response assay. The focus was on improving the text related to adhesion rather than remineralization throughout the document.

Differentiation of pulp stem cells. It is claimed, in the title and throughout  the ms., that EGCG induces odontoblast-like differentiation of pulp stem cells. This is not what is reported in the paper. Differentiation of pulp stem cells is induced by the differentiating medium; EGCG apparently magnifies the effect of the differentiating medium. To claim that EGCG induces differentiation, tests should be conducted with EGCG without differentiating medium. Again, the magnifying effect of EGCG is of interest, the claim is wrong.

Answer: Thanks for pointing this out. The overstatement of EGCG induce odontoblast-like differentiation was modified. The incorporation of EGCG as you mentioned, was the difference between differentiation mediums, the text was clarified.

The conclusions of the abstract and of the paper are, in this reviewer views, correct, reasonable and of interest: EGCG PROMOTES (this is different from INDUCES) odontoblast-like differentiation of pulp stem cells, possesses antibacterial effect, and increase SBS of dentin to the adhesive. The Authors should reword the ms. sticking to these sounds conclusions, not invoking mechanisms of phenomena, such as remineralization, that are not addressed in their experiments

Answer: The use of the term "induce" was replaced with "promote" throughout the document to accurately reflect the effect of EGCG and any content related to remineralization was revised and changed.

Round 2

Reviewer 3 Report

The ms. has been greetly improved, it can be accepted